# A Population Genomic Investigation of Immune Cell Diversity and Phagocytic Capacity in a Butterfly

**DOI:** 10.3390/genes12020279

**Published:** 2021-02-16

**Authors:** Naomi L. P. Keehnen, Lisa Fors, Peter Järver, Anna-Lena Spetz, Sören Nylin, Ulrich Theopold, Christopher W. Wheat

**Affiliations:** 1Department of Zoology, Stockholm University, Svante Arrheniusväg 18b, S-106 91 Stockholm, Sweden; fors.lisa@gmail.com (L.F.); soren.nylin@zoologi.su.se (S.N.); chris.wheat@zoologi.su.se (C.W.W.); 2Department of Molecular Biosciences, Wenner-Gren Institute, Stockholm University, Svante Arrheniusväg 20c, S-106 91 Stockholm, Sweden; peter.jarver@gmail.com (P.J.); anna-lena.spetz@su.se (A.-L.S.); uli.theopold@su.se (U.T.)

**Keywords:** phagocytosis, eco-immunology, functional genomics, innate immunity

## Abstract

Insects rely on their innate immune system to successfully mediate complex interactions with their internal microbiota, as well as the microbes present in the environment. Given the variation in microbes across habitats, the challenges to respond to them are likely to result in local adaptations in the immune system. Here we focus upon phagocytosis, a mechanism by which pathogens and foreign particles are engulfed in order to be contained, killed, and processed. We investigated the phenotypic and genetic variation related to phagocytosis in two allopatric populations of the butterfly *Pieris napi*. Populations were found to differ in their hemocyte composition and overall phagocytic capability, driven by the increased phagocytic propensity of each cell type. Yet, genes annotated to phagocytosis showed no large genomic signal of divergence. However, a gene set enrichment analysis on significantly divergent genes identified loci involved in glutamine metabolism, which recently have been linked to immune cell differentiation in mammals. Together these results suggest that heritable variation in phagocytic capacity arises via a quantitative trait architecture with variation in genes affecting the activation and/or differentiation of phagocytic cells, suggesting them as potential candidate genes underlying these phenotypic differences.

## 1. Introduction

Comparisons between populations have long been a powerful tool used by evolutionary biologists to identify both the action and potential targets of natural selection in wild populations (e.g., [1]). While common garden experiments can remove environmental effects to reveal genetic contributions to phenotypes, deeper dissection of phenotypic differences remains challenging. Phenotypic differences might arise from changes in the amount, size or performance of cells or organs. Disentangling these relative contributions to phenotypic differences is important, as the targets of selection are expected to be very different when the target is the genes directly involved in the phenotype vs. the proliferation or differentiation of cell types. Immune performance assays in natural populations often only focus upon phenotypic differences in immune performance without distinguishing between these two different causal routes (e.g., canonical immune genes vs. the development of cells involved in immunity). However, such insights are needed if we are to understand the microevolutionary dynamics of ecological immunity, as they inform upon whether the immune pathways or the development of immune cells are the target of selection on immune performance.

Insects are ubiquitous, occurring in a wide variety of environments. As such they are exposed to a wide variety of parasites and microbes, both from their environment as well as within themselves, i.e., their microbiome. These interactions with pathogens and parasites are mediated by their immune system. Natural variation in immunity is widespread among individuals [2], populations [3,4,5], and species [6]. This variation could be a response to geographic variation in biotic characteristics, e.g., bacterial density or parasite prevalence in the environment or within themselves [5,7,8,9], or abiotic factors such as temperature and nutrition [10,11,12,13]. These variable selection pressures can potentially shape and drive adaptations in the immune system [14,15,16,17,18].

The insect immune system can be divided into humoral and cellular defense responses. Humoral responses involve the production of antimicrobial peptides and enzymes of the phenoloxidase cascade [19]. The cellular responses, often the largest proportion of the immune response [20], are mediated by the blood cells (hemocytes), and include encapsulation, phagocytosis, and nodulation. Phagocytosis is one of the most widely conserved defense mechanisms against microorganisms. The process begins when pathogens or dead cells are recognized by pattern recognition receptors (PRRs), which bind the particle and activate intracellular cascades leading to the pathogen being engulfed into the phagosome, after which it is eliminated [21,22].

To our knowledge, studies on variation in phagocytic capabilities of invertebrates are scarce, but variation does exist within several evolutionary scales. At the macroevolutionary scale, comparative analysis across six arthropods found variation in phagocytic performance that was correlated with body size, as organisms with fewer hemocytes were smaller, yet had higher phagocytic capabilities [23]. Studies of closely related species have also been found to differ in their phagocytic activity, as two closely related heliothine moth species, *Heliothis irescens* and *H. subflexa*, differ in their phagocytic activity, hypothesized to be a result of their hemocyte composition differences [24]. In addition, phagocytic performance differences have been observed among different castes of the eusocial honeybee (*Apis mellifera*) [25]. Lastly, differences between sexes have also been observed in the scorpionfly (*Panorpa vulgaris)*, with females found better at phagocytosing than males [2]. However, we have been unable to find studies investigating population-level differences in phagocytosis.

While the literature at the phenotypic level is sparse, genome signatures of selection have identified genes having functional annotations related to phagocytosis. Phagocytosis-associated genes, across a wide variety of insects, show fast rates of evolution and high population differentiation [26,27,28,29]. In *Drosophila*, genes encoding PRRs specific to phagocytosis appear to evolve more rapidly and display high levels of population differentiation [26,30,31]. However, whether it was selection upon the phagocytic performance of these genes that drove this increased rate of evolutionary change remains to be investigated. In sum, while there is evidence for both phenotypic and genotypic variation in the wild in relation to phagocytosis, little is known about how this genetic variation interacts with variation on the phenotypic level, and whether this affects phagocytic capabilities. Furthermore, to our knowledge, the majority of the studies looking at phagocytosis have been performed at the species level, reflecting a lack of information obtained from spatial variability of immune defenses within species and between populations in the wild.

Here we work to integrate these top-down (phenotype level) and bottom-up (genomic patterns) perspectives by investigating the phenotypic and genetic variation related to phagocytosis in populations of a Lepidopteran butterfly, the green-veined white (*Pieris napi*). We aim to characterize differences between the populations in their innate immune responses, and then assess whether these correlate with divergence in relevant immune genes between these populations. Specifically, we investigate phagocytic performance variation at both the phenotypic and genomic level between two populations of the green-veined white butterfly (*P. napi*), one from Aiguamolls in northern Spain, the other from Abisko in northern Sweden. These are dramatically different habitats, with one above the polar circle and the other along the Mediterranean coast. *P. napi* is a common and widespread butterfly that feeds exclusively upon Brassicales plants, with limited dispersal and therefore low gene flow among populations [32]. This species produces between one (northern Europe) to four generations (southern Europe) per year depending on local conditions, reflecting local adaptation along latitude [33]. This variation in life history makes it an interesting species to investigate potential local adaptation of the immune system. Using common garden-reared individuals and several different measures of phagocytic performance, we find significant differences in the phagocytic performance between the two populations, as well as between sexes. Furthermore, we characterized the phagocyting cell types, and their relative performance, finding that both cell composition and propensity to phagocytose differ between the two populations. Finally, using a population genomics analysis, we investigated to what extent these populations differ in genes of the recognized phagocytic pathway, finding that there is little divergence or turnover at these genes. Instead, among genes exhibiting significant genomic divergence between these populations, genes involved in proteolysis, metabolic, and catabolic processes are identified as potential candidates for future study.

## 2. Materials and Methods

### 2.1. Study Species

*P. napi* adult females were caught in spring 2016 from Spain (Parc dels Aiguamolls de l’Empordà, north-east of Barcelona, 42.23° N, 3.10° E) and northern Sweden (Abisko 68.36° N, 18.79° E and Kiruna, 67.87° N, 20.17° E). Butterflies were kept separately in 1 L cups to lay eggs on *Alliaria petiolata*. Plants were replaced daily. Offspring were reared under 8:16 h light:dark, at 17 °C to ensure diapause development in both populations [33], and placed into 2 °C 0:24 Light:Dark for overwintering. They were kept per 4 in 1 L cups, and fed a mixture of wild *A. petiolata* and *Armoracia rusticana*. In spring 2017 the diapausing pupae were taken out of the cold, and reared for another generation as described above, only crossing between unrelated families. Larvae were observed during molting stages, and once they reached the final, fifth instar they were weighed and assayed.

### 2.2. Total Hemocyte Count

For the total hemocyte count (THC) assay, 20 Spanish (11F:9M) and 18 Swedish (10F:8M) fifth instar larvae were used. All larvae were sampled on the first day of the fifth instar, of approximately similar weight (weighed at a maximum of 2 h prior to the experiment). The larvae were bled by cutting the second proleg and 8 µL hemolymph was directly collected and added to 152 µL anticoagulant buffer (dilution 1:20), prepared according to [34] (62 mM NaCl, 100 mM glucose, 10 mM EDTA, 30 mM trisodium citrate and 26 mM citric acid). Samples were gently mixed and 10 µL was directly transferred to each side of a hemocytometer (Neubauer chamber, 0.1 mm depth). Hemocytes were counted in 10 squares for each individual and the cell concentration in 1 µl of hemolymph was calculated by dividing (the number of cells ×10) by (the number of squares × dilution).

### 2.3. Phagocytosis Slide Assay

To investigate the presence of phagocytic cells, a 10 µL suspension of heat-killed *Escherichia coli* (Thermofisher Scientific *E. coli* (K-12 strain) BioParticles™-Alexa Fluor™ 488 conjugate, a green-fluorescent dye, Invitrogen, Thermo Scientific, Waltham, MA, USA) was injected into early fifth instar larvae (22 larva per population, 11 of each sex). The bacteria were injected on the ventral side behind the second proleg using a glass capillary needle. After injection, the larvae were kept in separate containers with *Alliaria petiolate* ad libitum for 4 h (at room temperature (RT)) before sample preparation. After 4 h of incubation, the larvae were bled by cutting the second proleg, whereupon 5 µL of hemolymph per individual was collected and added to separate vials containing 300 µL phosphate-buffered saline (PBS) mixed with a small amount of phenylthiourea (PTU, Sigma Aldrich) and 5% newborn calf serum (NBCS, Biowest, Riverside, MI, USA). Individuals showing signs of open wounds were excluded to avoid the risk of unequal bacteria concentration. Two hemocyte samples were prepared from each individual larva. For each sample, a 30 µL drop of the mixture (hemolymph + PBS) was placed on a multi-spot microscope slide (SM-011, Hendley, Loughton, UK) and left to adhere for 20 min in a humid chamber at room temperature (RT). After adhesion, excessive hemolymph was gently removed with a pipette and the remaining cell monolayer was fixed with 4% PFA + PBS for 10 min. After fixation, cells were washed three times with PBS for 5 min. To distinguish morphological features and reveal the nuclei, the cells were treated with Phalloidin Rhodamine (Biotium, 3:1000 dilution in PBS) together with blue-fluorescent nucleic acid stain DAPI (Sigma-Aldrich, 1:1000 dilution in PBS) for 10 min in darkness, RT. After DNA-staining the cells were washed three times with PBS for 5 min and mounted in Fluoromount-G (SouthernBiotech, Oxfordshire, UK) mounting media. All samples were studied in a Zeiss Axioplan2, phase contrast, epifluorescent microscope connected to a Hamamatsu camera with Axio Vision 4.6. Eight random images per individual were taken and used for differential hemocyte counts and assessment of phagocytic capacity. Additionally, most samples were also studied in a Zeiss LSM 780 confocal microscope for acquisition of Z-stack images to further verify bacterial uptake. We determined two indices: i) the proportion of the different hemocyte types containing fluorescent bacteria (phagocytic index) (e.g., proportion of phagocytic granulocytes of total amount of phagocytes) and ii) the propensity of a given cell type to phagocytose, i.e., what is the proportion of a specific hemocyte cell type actively engaging in phagocytosis (e.g., the amount of phagocytic granulocytes of total amount of granulocytes), further mentioned as specific phagocytic index.

### 2.4. Quenching with Trypan Blue

To verify that the Alexa Fluor 488, heat-killed *E. coli* was ingested by the hemocytes and not just attached to the surface of cells, additional tests were performed using trypan blue to quench the signal from extracellular bacteria. Early fifth instar larvae were bled and 10 µL hemolymph per individual was collected and added to vials with 300 µL PBS mixed with 5% NBCS and a small amount of PTU as above. Then, a 2.5 µL suspension of *E. coli* was added to each vial and the solution was mixed gently but thoroughly, whereupon 30 µL of the mixture was placed on a multi-spot microscope slide and left to adhere for 20 min in a humid chamber at room temperature. Similar samples were prepared from injected individuals (without the addition of *E. coli* after bleeding). Just before the samples were studied in the microscope, 20 µL of each sample was removed and 10 µL freshly prepared 0, 4% trypan blue was added. After 1–2 min incubation, 10–15 µL of the mixture was removed with a pipette and a coverslip gently placed on top of the samples.

### 2.5. Phagocytosis Flow Cytometry Assay

To assess phagocytic capacity, in vitro samples of hemocytes treated with pHrodo-conjugated *E. coli* (pHrodo™ Green *E. coli* BioParticles^®^ Conjugate, Invitrogen, Thermo Scientific, Waltham, MA, USA) were prepared. The pHrodo Green conjugates are non-fluorescent outside the cell but fluoresce brightly green as pH decreases from neutral to acidic in the phagosomes, thereby providing a robust quantitative measure of phagocytic activity. A total of 74 Spanish and 48 Swedish larvae were used. All larvae used were in early fifth instar and were of approximately the same size (weighed at a maximum of 2 h prior to the experiment). The larvae were bled by cutting the second proleg, whereupon 25 µL hemolymph per individual was collected and added to separate vials containing 300 µL phosphate-buffered saline (PBS) with 5% newborn calf serum (NBCS, Biowest). The samples were then gently mixed with 50 µL pHrodo conjugated *E. coli* and incubated for 3 h (darkness, RT). Two control samples were prepared for each trial (one male and one female), and were treated with the same amount of *E. coli* but incubated on ice to inhibit the phagocytic process. After incubation, each sample was analyzed individually by a Fortessa flow cytometer (BD Biosciences, San Jose, CA, USA) and the data analysis was performed with FlowJo software (TreeStar, version 10.6).

### 2.6. Immunostaining of Hemocytes with Prophenoloxidase Antiserum

To distinguish the oenocytoids from other cell types, an additional test was performed using an antiserum to *Manduca sexta* prophenoloxidase (PPO; courtesy of Micheal R Kanost, Department of Biochemistry and Molecular Biophysics Kansas State University). Fifth instar larvae were bled by cutting the second proleg, whereupon 10 µL hemolymph per individual was collected and added to separate vials containing 300 µL phosphate-buffered saline (PBS) with 5% newborn calf serum (NBCS, Biowest). Three hemocyte samples were prepared from each individual larva. For each sample, a 30 µL drop of the mixture (hemolymph + PBS) was placed on a category number 1.5 coverslip (0.17 mm) and left to adhere for 1 h in a humid chamber at room temperature (RT). After adhesion, excessive hemolymph was gently removed with a pipette and the remaining cell monolayer was fixed with 4% PFA + PBS for 5 min. After fixation, cells were washed three times with PBS for 5 min, permeabilized with 0.1% Triton X-100 in PBS for 5 min and then washed again three times with PBS. Blocking was performed by adding 3% bovine serum albumin (BSA) for 30 min, after which the cells were incubated with the primary antibody (prophenoloxidase antiserum) for 2 h (darkness, RT). After washing with PBS three times, the cells were incubated with the secondary antibody (Alexa 488, anti-rabbit) for 2 h (darkness, RT), and then washed as above. To distinguish morphological features and reveal the nuclei the cells were treated with Phalloidin Rhodamine (Biotium, 3:1000 dilution in PBS) together with blue-fluorescent nucleic acid stain DAPI (Sigma-Aldrich, 1:1000 dilution in PBS) for 10 min (darkness, RT). After staining, the cells were washed four times with PBS for 5 min and mounted in Fluoromount-G (SouthernBiotech, Oxfordshire, UK) mounting media on a regular microscope slide. All samples were studied in a Zeiss LSM 780 confocal microscope.

### 2.7. Data Analysis

All statistical tests were performed using the statistical software R version 3.4.3. In order to correct for the pseudoreplicated nature of the data, generalized linear mixed models (GLM; lme4 package) were used, including the replicates as random factor. We performed stepwise backwards elimination for each GLM, starting with all interactions, and removing each least-significant term until the AIC (Akaike’s Information Criteria) and BIC (Bayesian Information Criterion) no longer changed. In the analysis of the propensity of a given cell type that was phagocytosing, not all slides contained all types of cells, so for these analyses we performed a GLM using quasibinomial distribution.

### 2.8. Gene Scan and Population Genetics

For this study the chromosomal level genome of *P. napi* was used [35]. As this genome, as well as other Lepidopteran genomes, have few annotations for phagocytosis genes, we generated annotations using proteins previously indicated to be involved with phagocytosis in *Drosophila melanogaster* (*N* = 155; [36]) using SPALN v2.1.2 [37,38]. To validate the accuracy of the identified phagocytosis genes, we implemented a number of validation steps. First, we used TBLASTN [39] with the 155 proteins to search the genome assembly, extracting hits with *E*-values < 0.0001 and a bitscore higher than 60 Sec, and we used MESPA [40] to conduct protein searches of the genome for the 155 candidate genes and generate gene models (*n* = 73 were predicted). We then compared the blast hits with the SPALN gene models, to assess gene model accuracy, filter out potential errors, and identify potential duplicates. As a last quality control step, each gene model was manually investigated using IGV [41]. To be confident in our gene models, average read depth per exon was calculated using coverageBed from Bedtools2 [42] compared to the coverage of all genes. These approaches resulted in a total of 73 phagocytosis-related genes identified in the genome.

For our population genomics analysis, we used two previously published [43] Pool-seq datasets originating from the two populations used in this experiment. These two libraries were mapped against the genome using Next-Gen mapper v.04.10. Samtools v1.2 was used to filter the mapped data for mapped paired-end reads, after which a mpileup file was created for further analysis [44]. Popoolation v1.2.2 [45] was used to mask indels using a 5-bp window, centered in the indel (identify-genomic-indel-regions.pl), as well as to mask non-genic regions (create-genewise-sync.pl). Population differentiation was identified using the fixation index (F_ST_), a measure of population differentiation wherein a 0 indicates no differentiation, whereas a value of 1 implies populations have fixed alternative allelic states and are completely differentiated. F_ST_ was calculated within the coding regions of genes, at the single nucleotide polymorphism (SNP) level, as well as estimated among all exons per gene, using Popoolation2 v1.201 [46]. In addition, F_ST_ was calculated for SNPs located within 5kb up and downstream of genes selected using the slopBed function of bedtools2 [42]. To assess whether the F_ST_ patterns of phagocytosis genes differed from the rest of the genome, two permutation subset analyses were performed in R [47] to control for differences in subset size. First, we estimated the mean F_ST_ of our 73 phagocytosis genes, and then compared this to a permutation test of 10,000 sets of 73 randomly sampled genome-wide genes. Secondly, we estimated the F_ST_ of 10,000 sets of 1906 randomly sampled genome-wide SNPs and compared this distribution of means to the mean of the 1906 phagocytosis SNPs. Finally, a GSEA was performed, and the SNPs with outlier values, i.e., SNPs with a value that that corresponded to the genome-wide 97.5th percentile, were used in GO-term enrichment analysis using TopGO v.2.36.0 [48]. We also added all of our identified phagocytosis genes to a phagocytic GO term for analysis. In topGO, the nodeSize parameter was set to 5 to remove GO terms which have fewer than five annotated genes, and other parameters were run on default.

## 3. Results

### 3.1. Total Hemocyte Count, Cell Type Identification and Characterization

In naive animals, the total hemocyte assay revealed that on average, a fifth instar *P. napi* larvae had 14,037 (SD = 7,125) hemocytes. There was no significant difference in total hemocyte count between larvae from different populations or sex (Figure 1a; GLM: P_population_ = 0.43, P_sex_ = 0.45, Appendix A). However, among animals injected with bacteria for our phagocytosis assay, the total number of hemocytes differed significantly between sexes, with females having a higher number of hemocytes than males (Figure 1b; *p* = 0.013); but, there was no significant difference between the populations (*p* = 0.092, Appendix A). Thus, while naive animals do not differ in total hemocyte counts, differences between the sexes were apparent after immune challenge.

During our experiments, three cell types were identified to be capable of phagocytosis and are therefore discussed more in depth. These were granulocytes, plasmatocytes, and (early) oenocytoids (Appendix A). The majority of the circulating hemocytes were granulocytes. Granulocytes are small, round cells with granular morphology and a centralized nucleus. Morphologically they can be identified by their protruding spikes from the regular circle and they have a weaker phalloidin staining than plasmatocytes (Figure 2A,B). Plasmatocytes are larger cells having a large elliptic nucleus, with long spikes that protrude in irregular shapes (Figure 2A). They often had weaker nuclear staining compared to granulocytes, but strong phalloidin staining of filaments. Oenocytoids are large, round or slightly elliptic cells with a slightly larger, often not centralized, nucleus than granulocytes and a more homogeneous cytoplasm (Figure 2C). Furthermore, they appear to have a smoother surface compared to the other cell types. In our studies we observed oenocytoids of very different sizes, representing different stages of this cell type. Their phagocytic capacity was mainly detected in the smaller cells, i.e., early or immature oenocytoids (2E), but was not observed in the larger cells, i.e., mature oenocytoids. Oenocytoid identity was further verified by immunostaining with prophenoloxidase-specific antiserum (Figure 2F).

These three cell types differed in their proportions with respect to population and sex. The proportion of granulocytes did not differ between the populations (Figure 3a, *p* = 0.21) or between sexes (*p* = 0.063, Appendix A). However, the proportion of plasmatocytes was significantly higher in larvae from Spain (Figure 3b, *p* = 0.029), but was not different between sexes (*p* = 0.14, Appendix A). Finally, for oenocytoids, there was an interaction between sex and population (*p* = 0.049, Appendix A), as males from Sweden had a higher proportion compared to the other sex and population (Figure 3c).

### 3.2. Phagocytic Index

We then investigated the number of phagocytes (i.e., cells that engulfed bacteria) between the populations, using our phagocytic assay (slide assay), and found that they differed significantly (*p* = 0.0003, Appendix A). Swedish larvae had significantly higher proportion of phagocytes (80.45%) compared to the Spanish larvae (66.90%). Furthermore, males on average had a higher proportion of phagocytes compared to females (Figure 4a, *p* = 0.029, Appendix A). Using an independent phagocytic assay (flow cytometry; gating strategy and examples of staining are shown in Appendix A), a similar population pattern was found, i.e., the Swedish had a significantly higher proportion of phagocytes compared to the Spanish (*p* < 0.001, Appendix A). However, the sex differences were not observed using this method (Figure 4b, *p* = 0.89, Appendix A). Thus, two independent methods found significant differences between the two populations in the phagocytic capacity of challenged individuals using two independent methods. However, the sex difference was only detected using the phagocytic slide assay.

### 3.3. Phagocytic Index by Cell Types

Phagocytic activity was quantified for the three main cell types: granulocytes, plasmatocytes, and early oenocytoids. The cell composition of these phagocytes did not significantly differ between the populations (proportion of phagocytic granulocytes: *p* = 0.52, SM Table 9; plasmatocytes: *p* = 0.60, SM Table 10; oenocytoids: *p* = 0.06, Appendix A), although Swedish larvae appeared to have a trend towards a higher proportion of phagocytic oenocytoids compared to Spanish larvae (*p* = 0.06; Sweden M = 9.70% SD = 4.91, Spain M = 7.33% SD = 3.27), similar to our previous observation of moderate differences in overall oenocytoid levels between populations (Figure 3c). In contrast, the cell composition of these phagocytes differed significantly between the sexes, with females having a higher proportion of phagocytic granulocytes (Figure 5a; *p* = 0.013, Appendix A) and significantly fewer phagocytic plasmatocytes (Figure 5b; *p* = 0.0349, Appendix A), but not oenocytoids, although there was a trend towards a higher proportion in males (Figure 5c; *p* = 0.086, Appendix A).

Next, we investigated the specific phagocytic index (i.e., the propensity of individual cellular types to phagocytose, e.g., the amount of phagocytic granulocytes of total amount of granulocytes). All three cell types showed a significantly higher propensity to phagocytose in Swedish larvae compared to Spanish larvae (Figure 6; GLM: P_granulocytes_ =0.00347, Appendix A; P_plasmatocytes_ = 0.0018, Appendix A; P_oenocytoids_ < 0.001, Appendix A). In addition, males exhibited a higher propensity to phagocytose in all three cell types (GLM: P_granulocytes_ = 0.02442, Appendix A; P_plasmatocytes_ = 0.0192, Appendix A; P_oenocytoids_ = 0.002887, Appendix A).

### 3.4. Population Differentiation

Having observed a significant difference between our common garden-reared populations in their phagocytic capability (Figure 4 and Figure 6), we conclude this likely to be heritable, that is, a genetically based phenotypic difference between populations. To explore the potential genetic basis for these differences in phagocytic capability, we next investigated whether the genetic divergence between the two populations was significantly different in 73 phagocytosis genes compared to the rest of the genome. First, F_ST_ values were calculated for each SNP within the coding region of genes. Genome-wide, a total of 363,459 exonic SNPs were identified and they had an average F_ST_ of 0.049 (Appendix A), showing a low level of allele frequency difference, as expected for allopatric populations. The 73 genes involved with phagocytosis did not differ significantly from this genome-wide pattern, suggesting that allelic variations at these loci are not associated with the observed phenotypic differences (Appendix A). Next, F_ST_ was estimated at the per gene level (*N*= 13,692 genes) to investigate the overall divergence patterns between the populations. Again, the genome-wide divergence between the population was relatively low (mean F_ST_ = 0.049; Appendix A), and permutation tests revealed that the 73 phagocytosis genes were significantly less diverged compared to the other genes in the genome (mean F_ST_ = 0.038; *p* = 0.9962; Appendix A). Finally, to investigate whether divergence in nearby regulatory regions could harbor divergent regions, we next investigated the 5kb up and downstream from each gene, with F_ST_ calculated at both the level of individual SNPs within this flanking region, as well as a single estimate for the entire region (flanking + coding region). Again, the significantly diverged regions of either analysis did not include the identified phagocytosis genes (Figure 7).

We then shifted our focus to move away from a candidate gene approach to focus on whether there were any genome-wide patterns that may explain our observed phenotypic differences. To do this, we selected the top 2.5% outliers in our gene level F_ST_ analysis to ask whether there was any functional enrichment within this gene set compared to the rest of the genome. This gene set enrichment analysis (GSEA) revealed that F_ST_ outliers are enriched for genes involved in proteolysis, metabolic, and catabolic functional processes, while no enrichment was found for phagocytosis function per se (Appendix A). When conducting our GSEA using the flanking regions, a significant enrichment was detected for genes involved in glutamine metabolism (Appendix A).

## 4. Discussion

To date, few studies have explored the potential geographic variability in cellular immunity within species and between populations. Further, determining whether phagocytic capacity differs due to the overall amount of cell types, their relative composition, or the phagocytic propensity of different cell types has rarely been addressed. Here we investigated the hemocyte amount and composition, as well as phagocytic capacity, in two allopatric populations of the green-veined white butterfly. We found that while the populations differed little in hemocyte amount and composition, they did differ significantly in their phagocytic capacity, which appears to be driven by increased level of their cells’ propensity to phagocytose. However, genome-wide analysis of the divergence between these populations found no excess genetic differentiation in genes annotated to phagocytic capacity, suggesting that our observed population differences might arise from genes affecting the activation and/or transdifferentiation of cells, which currently lack sufficient functional annotation.

### 4.1. Species Level Patterns: Sex Differences

According to Bateman’s principle, females are expected to invest more in post-copulation survival, since their fitness tends to be limited by their longevity, whereas males invest more in mating success. As a result, females are expected to invest more in their immune system [49]. Our results seem to be partly in concordance with this theory. Although the total hemocyte count (THC) of unchallenged individuals did not differ between the sexes, the THC in females was significantly higher when inoculated with bacteria. This discrepancy could be a result of a sex difference in response to bacterial detection, wherein the females proliferate more hemocytes after infections compared to the males. However, in contrast, our data also reveal the opposite direction: males have more phagocytes (cells that engulfed bacteria) than females. This is consistent with previous studies in *P. napi*, wherein sex differences were observed in both directions when assessing a humoral response, as females had a higher encapsulation rate while males exhibited a higher phenoloxidase activity [50]. Together, these results paint a complex picture, as it appears that there are differences between the sexes in their immune performance, but its direction depends on the immune phenotype studied. We did not measure whether the cell proliferation observed in females points towards an unquantified immune response, e.g., wound healing or phenoloxidase production, and thus this remains an open question.

### 4.2. Population Level Variation: Hemocyte Composition, Phagocytic Capacity, & Propensity

The populations differed in their cellular immunity in several ways. First, although the THC was not significantly different between the populations, the composition of the hemocytes did differ significantly. Spanish larvae contained a higher proportion of plasmatocytes, while Swedish larvae showed a trend for higher oenocytoid levels. Granulocytes, and to a lesser extent plasmatocytes, are reported to be the hemocytes responsible for phagocytosis in Lepidoptera [51,52,53]. However, our results reveal that early oenocytoids phagocytose at similar propensities as plasmatocytes. To our knowledge, this is a novel finding for Lepidoptera, though oenocytoids have been implicated in phagocytosis in several Coleopteran species [54,55]. Oenocytoids are the functional equivalent of Drosophila crystal cells and the early oenocytoids we observe may originate in a manner similar to novel crystal cells in Drosophila larvae, through transdifferentiation from plasmatocytes and/or granulocytes [56,57,58,59]. Secondly, we found a significant difference between the two populations in both their overall phagocytic capability, as well as their specific phagocytic index, i.e., the propensity of individual cellular types to phagocytose. This could be a result of local adaptation to geographic variation in parasite type or load. Our two populations do represent dramatically different habitats, one above the polar circle, one on the Mediterranean coast. Geographic variation in parasite pressure for butterflies has been observed before, as among adult monarchs captured at different points along the east coast fall migratory flyway, parasite prevalence declined as monarchs progressed southward [60]. Although geographic variation in parasite pressure is common among insect species, no data are available of the pathogenic environments of these two populations, and further research is needed to map this potential diversity to document and identify those of relevance to local adaption. Our results from the common garden experiments design, while suggestive of local adaptation, strongly suggest a genetic basis for the observed phagocytic difference between populations. To assess this, we investigated the divergence between these populations in the genes directly annotated to be involved in phagocytosis.

### 4.3. Population Genomic Analysis

Selection for differences in the immune performance between populations could act on several levels: the gene products (protein products), amount of products (protein levels), the cells that produce these proteins (cell proliferation), or the activation of cellular activity (cellular propensity). This selection can occur via acting on few alleles at a few loci with consequently large effect sizes, generating a large shift in allele frequency and subsequently reduced nucleotide diversity at these loci. Alternatively, selection could occur at many loci, each with small effect sizes that act additively, generating a more diffuse signal of allele frequency change at these loci. Despite observing a difference in phagocytic performance, genes previously characterized to be directly involved with phagocytosis (protein products) revealed no significant signal of genetic differentiation compared to the rest of the genome. Rather than selection acting directly upon the genes involved in phagocytosis itself, an alternative is that the phenotypic variation observed between populations has arisen through differences affecting the regulation, i.e., protein levels, of genes involved in phagocytosis. However, we found no excess divergence in the 5 kb flanking regions of phagocytosis genes. While differences in cell proliferation could give rise to our observations, our measures of THC and cell composition suggest this is not the case. Rather, our results may indicate that the selection has targeted either genes affecting the activation of phagocytic activity for each cell type, or a gene responsible for transdifferentiating cells, for example, plasmatocytes to oenocytoids, perhaps via a modulator gene. Unfortunately, given the lack of gene annotations for these functions, this hypothesis remains untested at this time.

A further alternative is that the genomic architecture is polygenic in nature rather than oligogenic, which would be difficult to detect given the diffuse nature of allelic changes that would then underlie our phenotypic changes between populations. One way to detect such a diffuse signal would be to query the genome for subtle allele frequency changes. To investigate this, we conducted a GSEA both at the gene and flanking regions, using a more relaxed measure of allele frequency change between populations. While the gene level analysis showed general GO terms involved with many organismal processes (Appendix A), more intriguing were the results from our GSEA of flanking regions, which revealed differences in genes involved in glutamine metabolism (Appendix A). Along with the Warburg effect, glutamine metabolic activities are increasingly implicated in activation of cells of both innate and adaptive immunity in mammals, including the differentiation of macrophages into M1 and M2 subtypes [61,62]. One may speculate that similar metabolic changes contribute to transdifferentiation between hemocyte classes in insects although [63], and while not mutually exclusive, they are also likely to be crucial for additional physiological adjustments. In sum, our attempts to connect bottom-up and top-down approaches supports the absence of large effect alleles at canonical genes involved in phagocytosis as underlying our observed differences in phagocytosis. Rather, we found a more diffuse signal among many of the genes that show significant allele frequency changes between our populations, which could be interpreted as informing upon our phenotype.

### 4.4. Pleiotropic Effects and Phagocytosis

The interaction between immunity and other aspects of physiology suggests that natural selection on a trait might exert indirect pressure on other correlated fitness traits. Resistance to infection is often considered to be a result of the immune system, however, resistance involves the entire physiology of the host [13]. *P. napi* has an adaptive cline across latitude in several traits, one of which is the number of generations they produce in a year. The longer and warmer growing season in Spain permits up to four generations per year, whereas the shorter growing season in northern Sweden only allows one generation, and individuals spend the majority of their lifespan as diapausing (i.e., overwintering) pupae [64,65]. Overwintering insects face pathogen and parasite pressures that change with the seasons, and as a result are expected to invest more in their overall immunity [66]. Secondly, immunity, longevity, reproduction, and metabolism are linked in a complex network via shared hormonal regulation [67] and insects that have a relatively long life are expected to invest in a more long-lasting sturdier body with more effective immune system [50,68]. Therefore, several indirect co-varying traits could be the basis of the variation in phagocytic capability found (i.e., the phagocytic propensity differences may arise indirectly via pleiotropy).

### 4.5. Hemocyte and Phagocytosis Insights from Multiple Angles

Here we measured hemocyte diversity and phagocytic capacity using several different approaches. Sometimes these agreed, other times they did not. The sex-linked difference in THC was found using cell slides, but was not detected by flow cytometry. This could be either due to the sample size used during the flow cytometer, or perhaps the type of cell that drives this difference is destroyed during this method. In sum, both methods have their strengths and potential biases. However, both methods are concordant in finding differences between populations in phagocytic capacity, while the sex-level differences were unique to the slide assay, consistent with the latter being perhaps a smaller effect. A final consideration are the candidate genes used in our study, as these were identified via homology with *Drosophila* to be involved with phagocytosis. *P. napi* lacks experimental studies confirming their direct role of these genes, which is unfortunately a limitation common in non-model organisms.

## 5. Conclusions

In sum, variation in the immune performance between populations can arise in various ways. We found evidence of differences in the phagocytic capability of populations, as well as the composition of the hemolymph. Cell types previously described to not be involved with phagocytosis, appear to have evolved the ability in our species. Our results suggest that to investigate the genetic basis of phenotypic differences in immunity, candidate gene approaches can be limited in their insights, calling for the need for genome-wide, unbiased studies, perhaps using QTL mapping or GWAS approaches, which are more sensitive to polygenic phenotype architectures.

## Figures and Tables

**Figure 1 genes-12-00279-f001:**
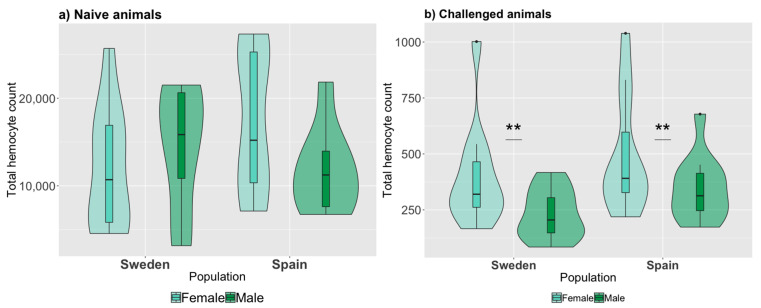
Total hemocyte counts for naive and challenged *Pieris napi* larvae. Total counts are shown (Y axis) per population and sex for (**a**) naive *P. napi* larva (*N* = 38 larvae) and (**b**) during the phagocytosis slide assay (*N* = 44 larvae). Bars and asterisks (**: *p* ≤ 0.01) indicate significant differences between sexes (population differences were not significant).

**Figure 2 genes-12-00279-f002:**
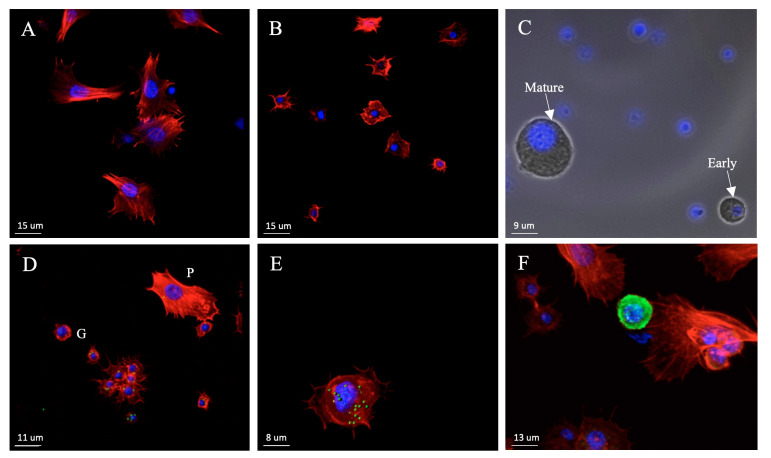
Morphology of the three main types of hemocytes in fifth instar *P. napi* larvae: (**A**) plasmatocytes, (**B**) granulocytes, and (**C**) early and mature oenocytoids. Hemocytes of *P. napi* larvae after in vivo phagocytosis of *E. coli* (AlexaFluor488, green): (**D**) plasmatocyte (indicated by ^P^), granulocytes (indicated by ^G^), (**E**) and early oenocytoid. (**F**) Staining of an oenocytoid with PPO antibody (green). Cell nuclei are stained with DAPI (blue) and the cytoskeleton of (**A**,**B**,**D**–**F**) with phalloidin (red). Scalebar = 15 um. Images taken using epifluorescent microscope (**C**) and confocal microscope (**A**,**B**,**D**–**F**).

**Figure 3 genes-12-00279-f003:**
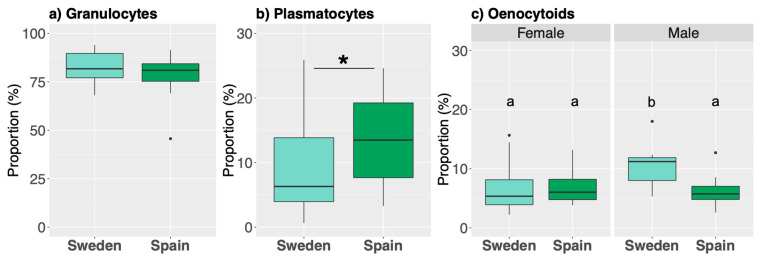
The overall cell composition of the total number of hemocytes during the phagocytosis assay as measured via imaging (*N* = 44 larvae). The most abundant cell types are granulocytes, followed by plasmatocytes and oenocytoids (panels **a**, **b**, and **c** respectively). Populations were only different for plasmatocye proportions, while only males from Sweden had a significantly higher proportion of oenocytoids. Within each bar plot, colored boxes contain X of the data, with whiskers representing the largest and smallest values within 1.5* interquartile range. The bar with asterisks (*: *p* ≤ 0.05) in panel b represents a significant difference between populations, while the letters above each bar plot in panel C denote significance relationships among these four samples of oenocytoids.

**Figure 4 genes-12-00279-f004:**
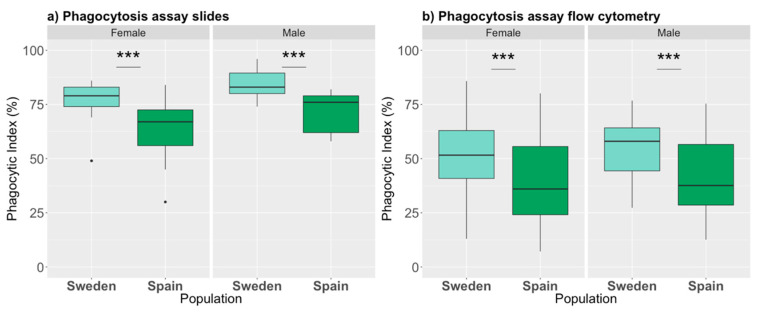
Proportion of phagocytes in *P. napi* larvae using two different assays. Phagocytic index (**a**) after injection of heat-killed *E. coli*, calculated via imaging (*N* = 44), and (**b**) after exposure to pHrodo-conjugated E. coli, detected by flow cytometry (*N* = 122). Bar plots are as described previously. Asterisks denote significance (***: *p* ≤ 0.001); note that for the phagocytosis slides males and females also significantly differed.

**Figure 5 genes-12-00279-f005:**
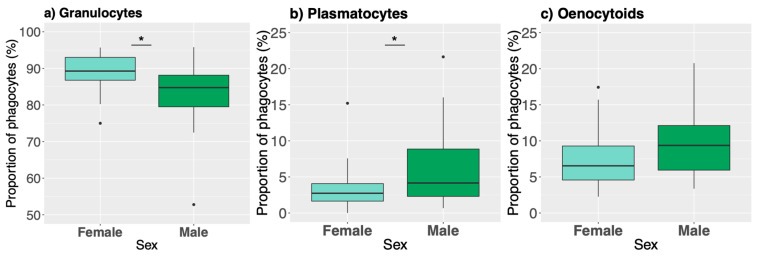
Differences between the sexes in cell composition of the phagocytes as measured via imaging (*N* = 44). (**a**) The majority of the phagocytes were granulocytes. (**b**) Males had significantly more phagocytic plasmatocytes compared to females. (**c**) Males showed a trend towards a highe proportion of phagocytic oenocytoids. Bar plots are as described previously. Asterisks denote significance (*: *p* ≤ 0.05).

**Figure 6 genes-12-00279-f006:**
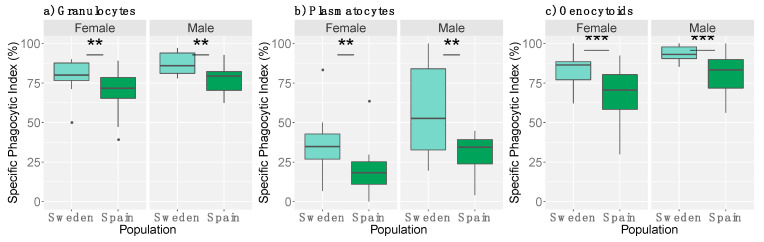
The specific phagocytic index of (**a**) granulocytes, (**b**) plasmatocytes, and (**c**) oenocytoids as measured via imaging (*N* = 44). Populations differed in their cells’ propensity to phagocytose. In addition, the sexes also differed (*p* < 0.05 *). Bar plots are as described previously. Asterisks denote significance (**: *p* ≤ 0.01; ***: *p* ≤ 0.001). Note the differences in y-axis.

**Figure 7 genes-12-00279-f007:**
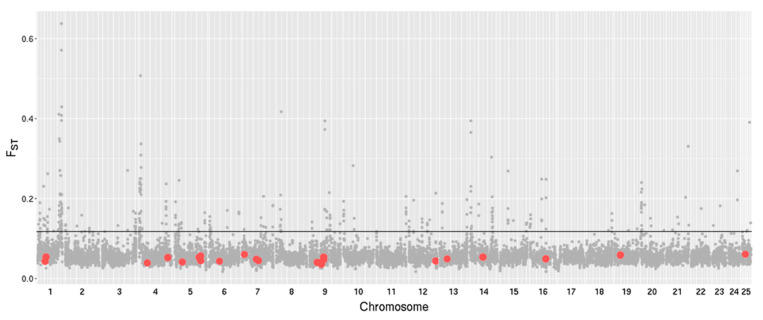
The genome-wide F_ST_ (fixation index) values of 5 kb flanking regions, estimated per gene (*n* = 13,692). Phagocytosis genes (*n* = 73) are indicated (red dots), with many in close proximity. The grey line shows the 97.5th percentile of the analysis, peaks above this line denote highly differentiated SNPs.

## Data Availability

The code used during the analysis will be made available on Github upon acceptance of the paper.

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
