# Peer review of "A Population Genomic Investigation of Immune Cell Diversity and Phagocytic Capacity in a Butterfly"

_genes, 2021, doi:10.3390/genes12020279_

Round 1

Reviewer 1 Report

Keehnen and coauthors made an interesting study combining phenotypic characterization of cellular immunity in allopatric populations of P. napi with a genomic approach aimed to find signatures of selection on candidate genes. They did not find any signal of genetic differentiation on candidate genes and they run a genome-wide scan to explain phenotypic differences. Overall, the paper is scientifically sound and interesting to a broad range of readers, from insect pathologist to evolutionary geneticists. Although the paper is well written, I found some parts that are not clear and interrupt the narrative fluid. Also, there is an experimental part that needs better explanation for readers not directly working with cellular immunity responses.  I add my comments below:

Introduction

Paragraph line 1- 44. Although I understand the message, several sentences are badly written, e.g. “Consider physiological differences, where phenotype differences might arise from changes in the amount, size or performance of cells or organs.” or “Here we draw attention to this issue in the field of ecological immunity, as immune performance assays in natural populations often only focus upon phenotypic differences in immune performance without distinguishing between these two casual routes”

Results

I guess that figures 2A and 2B are inverted in the figure (Line 315-317: Figure 2A points to granulocytes and 2B to plasmatocytes)

Title of supplementary table 2 should be “Test Statistics for total number of hemocytes” instead of “Test Statistics for total number of Phagocytosing cells”

Title Figure 4b: flow cytometry

Paragraph line 375 – 387 is repeated at lines 394- 405 (I guess this second location is the correct one).

Paragraph 3.4

I am not an expert in cellular immunity, but I cannot understand the results contained in this paragraph, neither the way they have been obtained. Phagocytic activity pointed that there were no differences in the cell composition (i.e. proportion of granulocytes, plasmatocytes and oenocytoids that engulfed bacteria) between the populations. Then, results mention the specific phagocytic index as the proportion of a given cell type to phagocytose, and here there are differences between populations. Which are the differences between the phagocytic activity and the specific phagocytic index? What was exactly measured in this second analysis? Authors should discuss the discrepancies between these two results in the discussion.

Line 422: “…we conclude this to be a heritable, that is, genetically...” change to “…we conclude this likely to be a heritable, that is, genetically…”

Author Response

Agreed. intro line 1-44. We have rewritten some of the sentences to be more comprehensive.

Agreed. results figure 2. We have now fixed the correct reference to the figure.

Agreed. We have changed the title of supplement table 2 as suggested.

Agreed. We changed the title of figure 4. 

Agreed. We have removed the first instance of the paragraph.

Comment on p 3.4

Agreed, it has been a challenging concept to put into words. I have adjusted and hopefully clarified it in the methods(L180), results(L430) and discussion (L533). In short, the phagocytic index of granulocytes reflect the % phagocytic granulocytes of total amount of phagocytes whereas the specific phagocytic index of granulocytes reflect the % phagocytic granulocytes of total amount of granulocytes. I hope this is now clearer.

Line 422:

Agreed. We have changed the sentence as suggested.

Reviewer 2 Report

This study compared the immune cell diversity and their phagocytic capacity between two populations of the butterfly Pieris napi, which are distantly located in north (Sweden) and south (Spain). Well designed and well written. Despite the not so much difference in hemocyte numbers between the two populations (both in intact and bacteria-challenged individuals), phagocytic capacity of the hemocytes are significantly higher in Sweden than in Spain. Interestingly, analysis of nucleotide diversity using genomic data identified significantly divergent genes that are involved in glutamine metabolism, which is considered to the linked to immune cell differentiation in mammals. Very interesting and provoking. Originality is high.

Minor comments:

L137: (F:M) Should numbers be shown here?

L142: A Space between 62 and mM

Author contributions: X.X. should be replaced.

Author Response

We thank the reviewer for their comments.

L137: Yes, thank you! Adjusted in text

L142: added a space. Adjusted in text

Author contributions, replaced X.X with NLPK

Adjusted in text